# IL-4/IL-13 polarization of macrophages enhances Ebola virus glycoprotein-dependent infection

Kai J. Rogers, Bethany Brunton[¤], Laura Mallinger, Dana Bohan, Kristina M. Sevcik, Jing Chen, Natalie Ruggio[ID], Wendy Maury[ID]*

Department of Microbiology and Immunology, University of Iowa, Iowa City, IA United States of America

¤ Current address: Department of Molecular Medicine, Mayo Clinic, Rochester, MN
* wendy-maury@uiowa.edu

## Abstract

### Background

Ebolavirus (EBOV) outbreaks, while sporadic, cause tremendous morbidity and mortality. No therapeutics or vaccines are currently licensed; however, a vaccine has shown promise in clinical trials. A critical step towards development of effective therapeutics is a better understanding of factors that govern host susceptibility to this pathogen. As macrophages are an important cell population targeted during virus replication, we explore the effect of cytokine polarization on macrophage infection.

### Methods/Main findings

We utilized a BSL2 EBOV model virus, infectious, recombinant vesicular stomatitis virus encoding EBOV glycoprotein (GP) (rVSV/EBOV GP) in place of its native glycoprotein. Macrophages polarized towards a M2-like anti-inflammatory state by combined IL-4 and IL-13 treatment were more susceptible to rVSV/EBOV GP, but not to wild-type VSV (rVSV/G), suggesting that EBOV GP-dependent entry events were enhanced by these cytokines. Examination of RNA expression of known surface receptors that bind and internalize filoviruses demonstrated that IL-4/IL-13 stimulated expression of the C-type lectin receptor DC-SIGN in human macrophages and addition of the competitive inhibitor mannan abrogated IL-4/IL-13 enhanced infection. Two murine DC-SIGN-like family members, SIGNR3 and SIGNR5, were upregulated by IL-4/IL-13 in murine macrophages, but only SIGNR3 enhanced virus infection in a mannan-inhibited manner, suggesting that murine SIGNR3 plays a similar role to human DC-SIGN. *In vivo* IL-4/IL-13 administration significantly increased virus-mediated mortality in a mouse model and transfer of *ex vivo* IL-4/IL-13-treated murine peritoneal macrophages into the peritoneal cavity of mice enhanced pathogenesis.

### Significance

These studies highlight the ability of macrophage polarization to influence EBOV GP-dependent virus replication *in vivo* and *ex vivo*, with M2a polarization upregulating cell surface

**Data Availability Statement:** All relevant data are within the manuscript and its Supporting Information files.

**Funding:** The following grants supported these studies: US National Institutes of Health R01 AI077519 to WM, US National Institutes of Health T32 GM007337, US National Institutes of Health T32 GM 067795,US National Institutes of Health GM 007337 to JC and US National Institutes of Health T32 AI007553 to DB. The US National Institutes of Health website is: www.nih.gov. The funders had no role in study design, data collection and analysis, decision to publish, or preparation of the manuscript.

**Competing interests:** The authors have declared that no competing interests exist.

receptor expression and thereby enhancing virus replication. Our findings provide an increased understanding of the host factors in macrophages governing susceptibility to filoviruses and identify novel murine receptors mediating EBOV entry.

## Author summary

Ebola virus causes outbreaks in Central and West Africa, often resulting in high mortality rates. Macrophages are important cell targets for the virus, yet infection of these cells remains poorly understood. Here, we show that macrophages stimulated with the immunomodulatory cytokines IL-4 and IL-13 are significantly more susceptible than unstimulated cells to a model virus that expresses the Ebola virus glycoprotein. These cytokines increase virus entry by enhancing the expression of the cell surface receptors DC-SIGN in humans and SIGNR3 in mice. Blocking availability of those receptors reduced virus load. Consistent with an important role for macrophages during EBOV infection, reconstitution of mice with macrophages treated with IL-4 and IL-13 exacerbated virus pathogenesis. Our studies argue for the critical importance of the macrophages and their response to immunomodulatory cytokines in controlling the pathological consequences of Ebola virus glycoprotein-dependent infections and highlight an important aspect of filovirus/ macrophage interaction that if controlled could decrease virus pathogenesis.

## Introduction

The genus *Ebolavirus* is composed of five viruses: *Zaire ebolavirus* (EBOV), *Sudan ebolavirus*, *Bundibugyo ebolavirus*, *Tai Forest ebolavirus* and *Reston ebolavirus*. All but Reston are pathogenic to humans and endemic in Africa. This genus is one of two extant genera that compose the family *Filoviridae*, with *Marburgvirus* being the other. Ebola virus disease (EVD) in humans is difficult to diagnose as it results initially in nonspecific symptoms common to a number of infectious agents endemic to Africa. Symptoms ultimately progress to a severe hemorrhagic fever with case fatality rates ranging from 25–90% [1, 2]. Historically, EBOV has caused sporadic outbreaks in Central Africa; however, the 2013–2016 epidemic in Western Africa demonstrated the potential of viral spread to other regions of Africa [3]. Most recently, a smaller, but persistent EBOV outbreak is ongoing in the Democratic Republic of the Congo, highlighting the continued re-emergence of this pathogen. No targeted therapeutics have come to market, with little more than supportive care currently available to patients [4].

Virus entry is a target for antiviral development. Filoviruses enter susceptible cells by interactions with several different cell surface receptors that mediate virion attachment and internalization into the endosomal compartment. Two different groups of cell surface receptors are known to mediate filovirus uptake: phosphatidylserine (PS) receptors, such as the TIM and TAM family of proteins, and C-type lectin receptors (CLRS) that bind to glycans on the heavily glycosylated viral glycoprotein (GP) [5, 6]. To date, five different PS receptors and five CLRs have been shown to facilitate filovirion uptake [7–11]. Binding to these receptors mediates uptake of virions into endosomes where the filovirus GP is proteolytic processed [12, 13]. Cleaved GPs interact with the cognate receptor NPC1 within the late endosomal/lysosomal compartment [14]. While NPC1 binding is required for filovirus entry, studies suggest at least one additional step is required to trigger membrane fusion events [14, 15]. Given the redundancy of attachment/internalization receptors and ubiquitous expression of endosomal

proteases and NPC1, it is not surprising that a wide range of host cells are susceptible to infection, including macrophages.

Macrophages are known to play a critical role in EBOV pathogenesis. These cells are one of the initial cell types infected by EBOV and are targets throughout the course of infection, facilitating systemic dissemination of the virus [16–19]. Infection of macrophages results in aberrant production of proinflammatory cytokines, synthesis and secretion of vasoactive peptides disrupting vascular permeability, and recruitment of additional susceptible cells to the site of infection. This creates a positive feedback loop to allow further viral replication and systemic spread [20]. However, macrophages are not phenotypically homogenous; rather, they adopt a broad spectrum of phenotypes depending on their microenvironment, including the cytokines to which they are exposed [21]. Their remarkable phenotypic plasticity in response to cytokines and other stimuli is referred to as "macrophage polarization". M1 or classically activated macrophages are elicited by exposure to proinflammatory signals such as interferon gamma (IFN-γ). M2 or alternatively activated macrophages are immunomodulatory macrophages and elicited by signals such as the cytokines IL-4, IL-10, IL-13 and TGF-β, as well as additional immunosuppressive chemicals such as glucocorticoids or combination treatments of immune complexes (IC) and bacterial LPS [22]. Importantly these cells perform disparate functions in the body. M1 macrophages upregulate expression of a number of proinflammatory genes such as CXCL10, IL-12 subunits, IL-1β, TNF, and IL-6 and mediate clearance of pathogens. M2 macrophages produce a number of immunomodulatory compounds and promote resolution of inflammation and wound healing. More recently, it has become clear that the classification of M2 is insufficient to describe the unique and specific expression profiles induced by disparate anti-inflammatory stimuli. As a result the field has moved towards subdividing the M2 macrophage populations and frequently refer to the polarization by the specific stimulus administered to elicit the unique and characteristic macrophage phenotype [21]. Thus, IL-4/IL-13 treatment is referred to as IL-4/IL-13 or M2a polarization, whereas, IC/LPS or dexamethasone treatments are frequently referred as M2b or M2c polarization, respectively.

A growing body of research indicates that macrophage polarization may occur in various disease states and this may profoundly impact the outcome of the host immune responses [23]. Macrophage populations are skewed towards either M1- or M2-like phenotypes in many chronic disease states such as diabetes, atherosclerosis and cancer [24–27]. Further, cytokines that drive macrophage polarization state influence the outcomes of infection by such diverse pathogens as HIV-1, influenza virus, Salmonella spp., Brucella, Leishmania spp., and others [27–32].

Given the critical role of these cells in the course of filovirus infection, the polarization status of macrophages might well influence EBOV infection. Studies in our lab have demonstrated that IFN-γ treatment, resulting in M1 polarization of human or mouse macrophages, inhibits EBOV infection and in vivo IFN-γ administered as late as 24 hours following EBOV challenge abrogates morbidity and mortality [33]. These findings provide evidence that eliciting a proinflammatory state at early times during filovirus infection can control filovirus replication. However, tissue macrophages of many West and Central African individuals may not manifest a strong proinflammatory phenotype during early filovirus infection due to other infections. For example, helminth infections are common in this region and they induce a systemic M2/Th2 response, pushing the immune response towards an anti-inflammatory phenotype [34–38]. Further, while there is no evidence that filovirus infection directly pushes macrophages towards an M2-like state, it is known that filoviruses encode an IFN-γ antagonist that would be predicted to block effective M1 polarization [39, 40]. Thus, it is important to understand the effect of M2-like macrophage polarization on filovirus infection.

Investigations of the effect of immunomodulatory cytokines on EBOV infection of macrophages are limited, but suggest that M2-like polarization of macrophages enhances filovirus

infection. Maturation of monocytes into macrophages enhances Ebola virus like particle (EBOV VLP) entry and virus entry is increased modestly by M2-polarizing stimuli, suggesting that cytokine polarization alters the susceptibility of these cells for EBOV [41]. Also, the M2 stimulus, IL-10, exacerbates EBOV infection as IL-10 knockout mice have reduced EBOV load in some organs at late times during infection and have better survival than wild-type mice [42]. These studies support the hypothesis that anti-inflammatory agents may enhance EBOV infection; however, mechanism(s) mediating this enhancement have yet to be established.

Here we show that IL-4/IL-13 polarization of murine peritoneal macrophages enhances EBOV GP-dependent infection by increasing cell surface expression of CLRs. In these studies, we primarily use BSL-2 recombinant VSV encoding EBOV GP (rVSV/EBOV GP) as a cost-effective and safe surrogate model of EBOV infection. This recombinant virus has been used by numerous groups to study filovirus entry and offers the distinct advantage of allowing filo-virus entry studies to be performed outside of BSL-4 laboratories [14, 33, 41, 43, 44]. Administration of IL-4/IL-13 to macrophages significantly enhanced EBOV GP-dependent viral particle binding and internalization with increases in overall infection. IL-4/IL-13 also increased surface expression of several C-type lectin receptors (CLRs) on human monocyte derived macrophages or murine peritoneal macrophages, including DC-SIGN on human macrophages and SIGNR3 on mouse macrophages. Enhanced IL-4/IL-13-mediated infection was blocked by mannan, indicating that mannose-binding CLRs, such as DC-SIGN, are responsible for the increased virus infection. Ectopic expression of these receptors in poorly permissive cells increased rVSV/EBOV GP entry, but other related CLRs did not. IL-4/IL-13 that was administered to mice prior to rVSV/EBOV GP challenge increased virus-associated mortality. Further, transfer of *ex vivo* IL-4/IL-13-stimulated peritoneal macrophages to naïve mice enhanced rVSV/EBOV GP pathogenesis. In total, these studies demonstrate that IL-4/IL-13 polarization of macrophages generates a macrophage that is highly susceptible for EBOV entry and infection through enhancement of cell surface CLRs, and further highlight the importance of macrophages in the pathogenesis of EBOV.

## Methods

### Ethics statement

This study was conducted in strict accordance with the Animal Welfare Act and the recommendations in the Guide for the Care and Use of Laboratory Animals of the National Institutes of Health (University of Iowa (UI) Institutional Assurance Number: #A3021-01). All animal procedures were approved by the UI Institutional Animal Care and Use Committee (IACUC) which oversees the administration of the IACUC protocols and the study was performed in accordance with the IACUC guidelines (Protocol #8011280, Filovirus glycoprotein/cellular protein interactions).

### Mice

C57BL/6 IFN-α/β receptor-deficient (*Ifnar*$^{-/-}$) mice were a kind gift from Dr. John Harty, University of Iowa. C57BL/6 *Timd-4*$^{-/-}$ mice were a kind gift from Dr. Vijay Kuchroo (Brigham & Women's Hospital, Harvard Institute of Medicine). Mice were bred in our breeding colony. C57BL/6 *Ifnar*$^{-/-}$ and C57BL/6 *Timd-4*$^{-/-}$ mice were crossed to create heterozygous progeny. Progeny of the F1 generation were interbred and mice were screened for the correct double knockout (*Ifnar*$^{-/-}$/*Timd-4*$^{-/-}$) genotype. All expected genotypes were produced in normal Mendelian ratios. Genomic DNA from mouse tail clips was assessed by PCR for genotypes. *Timd-4* genotyping primers have been previously described [45] and were utilized in PCR amplification for 30 cycles at 94˚C for 30 sec, 55˚C for 30 sec, and 72˚C for 1 m.

Mice were maintained in accordance to IACUC guidelines at the University of Iowa. All infections were performed using a virus that provides predictable killing in mice (determined for each stock of virus prepared and highly dependent on the sex of the mouse). The dose used in individual experiments was chosen to best address the experimental question being asked. Infections were performed by intraperitoneal injection and mice were monitored daily for signs of morbidity. Mice were euthanized in accordance with defined IACUC protocols. To harvest organs, mice were anesthetized with isoflurane and perfused through the left ventricle with 10mL PBS to flush the vasculature. Organs were harvested, weighed, and snap frozen in liquid nitrogen. RNA was isolated from whole organs using TRIzol as previously described. For *ex vivo* experiments, peritoneal macrophages were isolated and polarized as described below. Following polarization, cells were washed vigorously in PBS before being lifted with Versene and washed in sterile PBS to remove any residual media/dissociation reagent. Adoptive transfer was subsequently performed by injecting $1.5 \times 10^6$ pmacs into each recipient mouse and infecting 24 hours following later.

## Primary cell isolation and macrophage polarization

Resident peritoneal macrophages were obtained from mice by peritoneal lavage with 10mL of RPMI + 1% pen/strep. Cells were washed once with PBS and resuspended in RPMI containing 10% FBS, 1% pen/strep, 1% non-essential amino acids (NEAA),1% sodium pyruvate, and murine MCSF at 20ng/mL. After 48 hours, cells were washed with PBS which removed most of the non-adherent cells. This generated a macrophage enriched population of cells composed of greater than 85% macrophages.

Bone marrow derived macrophages were obtained from bone marrow precursors isolated from the femurs of adult mice. Cells were obtained by flushing the lumen of the femur with RPMI containing 10% FBS and 1% pen/strep. Cells were passed through a 70μm cell strainer and washed with PBS before being plated in the isolation media containing 20ng/mL murine MCSF to promote differentiation to macrophages. Cells were ready for use following 6 days of differentiation.

Human macrophages were matured from monocytes obtained from leukocyte reduction cones containing enriched peripheral leukocytes from healthy donors at the DeGowin Blood Center at University of Iowa Hospitals and Clinics. Mononuclear cells were purified by Ficoll gradient, and monocytes were selected by adherence to tissue culture flasks coated in 2% gelatin and pre-treated with human plasma. Following isolation, monocytes were plated in RPMI with 10% FBS, 1% pen/strep, 10% human serum, and 20ng/mL human MCSF. Cells were matured for 6 days at which point they were washed with PBS to remove non-adherent cells. Cells were polarized with human cytokines on day 6 of differentiation.

Polarization of macrophage cultures was achieved by incubating cells for 24 h in the plating media without MCSF and with 20ng/mL IL-4 + 20ng/mL IL-13 for 24 hours. Following polarization, media was removed and replaced with the plating medium without cytokines and harvested for RNA or infected with virus.

## RNA isolation and qRT-PCR

RNA was isolated using the TRIzol reagent from Invitrogen. All steps were performed according to the manufacturer's specifications. RNA was subsequently converted to cDNA with the High Capacity cDNA RevTrans Kit (#4368814) from Applied Biosystems. A total of 1μg of RNA was used as input for each reaction. Quantitative PCR was performed using POWER SYBR Green Master Mix (#4367659) from Applied Biosystems according to the manufacturer's instructions and utilizing a 7300 real time PCR machine from Applied Biosystems. 20ng of

cDNA were used in each well. Primers used are as follows: HPRT forward 5'-GCGTCGTGAT TAGCGATGATG-3', HPRT reverse 5'-CTCGAGCAAGTCTTTCAGTCC-3', VSV-M forward 5'-CCTGGATTCTATCAGCCACTTC-3', VSV-M reverse 5'-TTGTTCGAGAGGCTGG AATTAG-3', GAPDH forward 5'-GGTGTGAACCATGAGAAGTATGA-3', GAPDH reverse 5'-GAGTCCTTCCACGATACCAAAG-3', DC-SIGN forward 5'-GAGTTCTGGACACTGG GGGA-3', DC-SIGN reverse 5'-TGGCCAAGACACCCTGCTAA-3', MGL forward 5'-CTTT GAGAAAGGCTTTAAGAACTGG-3', MGL reverse 5'-TCATCTCACAGATCCAGCGG-3', IRF-1 forward 5'-TCACACAGGCCGATACAAAG-3', IRF-1 reverse 5'-GATCCTTCACTTC CTCGATGTC-3', CD163 forward 5'-AGCGGCTTGCAGTTTCCTCA-3', CD163 reverse 5'-GACACAGAAATTAGTTCAGCAGCA-3', FIZZ-1 forward 5'-ACTTCTTGCCAATCCAGC TAAC-3', FIZZ-1 reverse 5'-CAAGCACACCCAGTAGCAGT-3', Arg1 forward 5'-TTT TAG GGT TAC GGC CGG TG-3',

Arg1 reverse 5'-CCT CGA GGC TGT CCT TTT GA-3', YM1 forward 5'-AAG CTC TCC AGA AGC AAT CCT-3',

YM1 reverse 5'-TAG GAA GAT CCC AGC TGT ACG-3', GBP5 forward 5'-CCC AGG AAG AGG CTG ATA G-3', GBP5 reverse 5'-TCT ACG GTG GTG GTT CAT TT-3', VAMP5 forward 5'-GCT CAA CAA TTT CGA CAA GGT C-3', VAMP5 reverse 5'-GGG CTA AAG TCT GGT TG TCT-3', NPC1 forward 5'-TGC CAC AGA AGG CGG TAC TT-3', NPC1 reverse 5'-TTG TGA ACA TGC GCC TCA GA-3', MerTK forward 5'-TCCGACTTCTAGGC GTGTGT-3', MerTK reverse 5'-GTTTCGAGCTGCCAAATCCC-3', TYRO3 forward 5'-CTG CTCCCTCTCCCTCTGTTT-3', TYRO3 reverse 5'-GCGTTGTAGCTGGAGATCGTT-3', Axl forward 5'-TGAGCCAACCGTGGAAAGAG-3', Axl reverse 5'-AGGCCACCTTATGCCGAT CTA-3', Tim-1 forward 5'-TCCCATCCCATACTCCTACAGA-3', Tim-1 reverse 5'-TAAGTA TGTACCTGGTGATAGCCAC-3', Tim-4 forward 5'-GGCTCCTTCTCACAAGAAACCA CA-3', Tim-4 reverse 5'-TCAGCTGTGAAGTGGATGGGAGA-3', Integrin alpha 5 forward 5'-TGATTCAACAGGCAATCGAGA-3', Integrin alpha 5 reverse 5'-CCGTCCTGAAGAAA GCAGGT-3', MSR1 forward 5'-CCAGCAATGACAAAAGAGATGACA-3', MSR1 reverse 5'-CTGAAGGGAGGGGCCATTTT-3', CD300a forward 5'-GGACCAACACTAGAGACAC CT-3', CD300a reverse 5'-GCCTATCAGCTTTGACCAGC-3', SIGNR1 forward 5'-GGGCTC CTGCTGATCATTCT-3', SIGNR1 reverse 5'-CCTGGACGTAAGCTCATCTGT-3', SIGNR3 forward 5'-CACCCAGCTTTACACGCTATTGG-3', SIGNR3 reverse 5'-GATGCAGAGGCT GAGATGAGG-3', SIGNR5 forward 5'-GCTGGCGTAGATCGACTGT-3', and SIGNR5 reverse 5'-ACAAGTTGAGCCCCCACATT-3'.

## Surface staining

All surface staining was performed in the presence of the Fc receptor blocker monoclonal antibody 2.4G2 (BioXCell) at 20μg/mL. The following antibodies were used to detect CLECs on the surface of cells: anti-murine CD209b (SIGNR1) clone eBio22D1 (Invitrogen), anti-murine CD209a (SIGNR5) clone LWC06 (ebioscience), anti-murine CD209d (SIGNR3) catalog number AF7220 (R&D Systems), anti-human DC-SIGN clone 1B10 (Santa Cruz Biotechnology), anti-murine TIM-4 catalog number AF2826 (R&D Systems), anti-murine TYRO3 catalog number AF759 (R&D Systems), anti-murine MGL1/2 catalog number FAB4297P (R&D Systems). Cells were stained in FACS tubes in the dark on ice for 20 minutes (an additional 15-minute stain for those requiring secondary antibodies). Two washes with PBS + 2% FBS were performed after each stain to reduce background. Cells were then run on either a BD FACSCalibur or BD FACSVerse flow cytometer and fluorescence was quantified by percent positive cells using FlowJo v10 (TreeStar).

## Viruses

Recombinant vesicular stomatitis virus expressing the glycoprotein from EBOV (Mayinga) or VSV G was generated as previously described [46]. Virus was propagated by infecting Vero cells at low MOI (~0.05) and collecting supernatants at 48hpi. The resulting supernatants were filtered through a 45-micron filter and purified by ultra-centrifugation (28,000g, 4˚C, 2 hr) through a 20% sucrose cushion. The resulting stocks were resuspended in a small volume of PBS and frozen until use. Those stocks used for *in vivo* studies were further purified by treatment with an endotoxin removal kit (Detoxi-Gel Endotoxin Removing Gel, ThermoFisher Scientific 20339) before being aliquoted and stored at -80˚C until use. All viral titers were determined by $TCID_{50}$ assay on Vero cells. An important note: multiple virus stocks were used through the course of these experiments. The relationship between the $TCID_{50}$ and $LD_{50}$ for each stock was unique for that stock. For each stock, the lowest dose of virus providing consistent lethality was determined and the amount of virus in each experiment was determined by whether sub-lethal or lethal doses of viruses was required for the study.

## Infections in tissue culture

The recombinant viruses, rVSV/EBOV GP and rVSV/G, were assessed for their ability to infect cytokine treated and untreated primary human and mouse macrophages as well as transiently transfected HEK 293T cells in these studies. Infectivity was assessed by two methods. As both viruses encoded GFP in their genomes, infectivity was assessed in some studies by the number of GFP+ cells in infected wells at 24 hours following infection. For these assays, cells were infected with a quantity of virus that achieved ~10–20% GFP positivity in unstimulated cultures. This level of virus infection is in the linear range of the infection curve, allowing detection of subtle increases and decreases in viral infectivity. Approximate MOIs used in each study are noted in the figure legends. Virus infections were also assessed by viral load determinations via qRT-PCR analysis using primers and conditions described above.

## Production of virus like particles

HEK 293T cells were seeded at $2x10^6$ in 10cm tissue culture plates and 10mL of DMEM + 10% FBS + 1% PS. Cells were co-transfected using an optimized PEI protocol with four separate plasmids: NP, VP40, VP40 containing β-lactamase, and EBOV GP at a ratio of 1:2.5:2.5:1 respectively. Approximately sixteen hours post-transfection, the media on the plates was replaced with 10mL of fresh media. Supernatants were then collected at 24 and 48 hours. Supernatants were filtered through 0.45μm 30mm PVDF syringe filter units (CellTreat cat# 229745). VLPs were concentrated overnight at 7,000 rpm and 4˚C. The virus was further purified by ultracentrifugation for 2hrs at 28,000 rpm and 10˚C through a 20% sucrose cushion in a SW 60Ti rotor. The purified viral particles were then thoroughly resuspended in PBS and aliquoted. To determine an approximate concentration of VLPs, the protein concentration was determined using BSA as a standard. VLPs were analyzed on a western blot after being stained for GP and VP40 (IBT Bioservices GP cat# 0200–001, VP40 cat# 0201–016, respectively) and GP signal was compared to serial dilutions of a purified aliquot of soluble GP (Sino Biologics).

## Transfections

Transfection of HEK 293T cells was performed using the reagent polyethylenimine (stock 1mg/mL). For transfection of a 10cm dish, 16μg of plasmid DNA in 400μl 150mM NaCl was combined with 48μl PEI in 352 μl NaCl. Mixture was incubated at room temperature for 20 minutes and added dropwise to cells. 24 hours following transfection, cells were washed with

PBS and re-plated in a 48 well dish in DMEM with 10% FBS. 24 hours after re-plating cells were ready for use as protein expression has been found to peak at 48 hours post transfection.

### Competitive inhibitors

Both mannan and N-acetylgalactosamine (GalNac) were obtained as powders from Fisher Scientific and resuspended to a concentration of 1mg/mL in sterile PBS. Competitive inhibition of virus binding was performed by incubating target cells with the inhibitors at a final concentration of 200μg/mL for one hour, and subsequently infecting in the continued presence of the inhibitor.

### Phosphatidylserine liposome competition studies

L-α-phosphatidylserine (PS) (Sigma) liposomes were generated by dissolving in chloroform, drying, resuspending in PBS, and sonicating for 5 minutes on ice as previously described [47]. PS liposomes were added to cells at a concentration of 50μM one hour prior to the addition of virus to allow for competitive binding to occur as previously shown [48].

### Quantification of virus binding and entry

Binding assays were performed by placing cells on ice and adding rVSV/EBOV GP (MOI = 1) for 1 hour to allow for binding but not internalization. Cells were subsequently washed twice with PBS to remove any unbound virus. Virus bound cells were lysed with TRizol and RNA was isolated, converted to cDNA, and viral cDNA was detected by quantitative PCR. Entry assays were performed utilizing the EBOV VLPs containing VP40-β-lactamase described above. The assay was performed as described by Tscherne and Garcia-Sastre [49]. Briefly, cells were incubated in Opti-MEM with EBOV VP40-BLac VLPs (approximately 100ng protein per well) for 1 hour on ice to allow binding. Cells were washed, resuspended in media and shifted to 37°C for 4 hours. Cells were then washed, lifted with trypsin, and moved to FACS tubes. Cells were resuspended in Opti-MEM containing the LiveBLAzer FRET reagent with CCF2-AM prepared according to manufacturer's guidelines (Invitrogen). Substrate was prepared according to manufacturer's guidelines. Cells were subsequently washed and evaluated by fluorescence in the V500A and Pacific Blue channels on a BD FACSVerse to quantify substrate cleavage.

### Statistics

All *in vitro* experiments are shown as mean with error expressed as standard deviation. Significance was determined by two tailed Student's t-test (alpha = 0.05). For *in vivo* experiments, significance was determined using Log-rank (Mantel-Cox) test. All statistics were calculated using GraphPad Prism software (GraphPad Software, Inc.).

## Results

### M2 polarization of macrophages increases susceptibility to rVSV/EBOV GP

To evaluate the role of macrophage polarization on EBOV infection, we utilized rVSV/EBOV GP, a virus that is an excellent model for studying EBOV entry events [13, 41, 43, 44, 50–53]. Since rVSV is strongly inhibited by type I interferons [54], murine monocytes and macrophages were obtained from interferon α/β receptor $^{-/-}$ (*Ifnar$^{-/-}$*) mice, however key *in vitro* experiments were also performed using WT murine cells and human monocyte derived macrophages (MDMs) which are intact for type I interferon signaling. Using these cells, we sought to determine the effect of M2-like polarization on virus infection.

Because IL-4/IL-13 or M2a polarization is shown to play a clear role in other disease states and is likely to be relevant to a variety of helminth-infected African patient populations exposed to EBOV [55], we focused here on the effect of M2a polarization on rVSV/EBOV GP infection. While a large number of studies have used IL-4 and IL-13 to elicit the M2a pheno-type in macrophages [31, 56, 57], we verified in our primary cells of interest, murine resident peritoneal macrophages (pmacs), that changes in gene expression occurred that were reflective of functional macrophage polarization following IL-4/-13 treatment. We found changes consistent with those reported by others, allowing us to confidently utilize these cytokines to induce an M2a phenotype [58, 59] (S1 Fig). We next investigated the effect of these cytokines on the ability of both murine pmacs and human MDMs to support infection of rVSV/EBOV GP. We found that IL-4/IL-13-stimulated macrophages were more robustly infected by rVSV/EBOV GP as compared to non-polarized controls, with a 3-4-fold increase in infection observed (Fig 1A and 1B). In contrast, infection with rVSV/G, expressing native VSV G glyco-protein, was not enhanced by IL-4/IL-13 stimulation, suggesting that the increase in infection is specific to EBOV glycoprotein-expressing virus. To verify that the effect observed in murine pmacs was not impacted by the lack of type 1 interferon signaling, we performed a similar experiment in WT pmacs and found that IL-4/IL-13 treatment increased infection 2.5-fold, consistent with the effect observed in *Ifnar$^{-/-}$* cells (Fig 1C).

## IL-4/IL-13 stimulation of macrophages enhances rVSV/EBOV GP entry

As our findings suggested that IL-4/IL-13 treatment enhanced rVSV/EBOV GP infection in an EBOV GP-dependent manner, we evaluated the earliest steps in infection: binding and entry. Pmacs or human MDMs were pretreated with IL-4/IL-13 for 24 hours and rVSV/EBOV GP was bound to cells at 4˚C to prevent virion internalization. Unbound virus was removed, total RNA was isolated, and the quantity of virus bound to the cells was assessed by qRT-PCR amplification of viral genomes. Virions bearing EBOV GP bound to IL-4/IL-13-treated macro-phages more efficiently than to non-polarized macrophages (Fig 2A and 2B), suggesting that M2 polarization enhanced surface expression of viral adhesion factors/receptors. To determine if virion fusion events were also enhanced in IL-4/IL-13-treated macrophages, EBOV virus like particles (VLPs) containing EBOV VP40 fused to β-lactamase [60] were assessed for fusion efficiency. EBOV VLP fusion events were measured in cells loaded with the fluorescent β-lac-tam containing dye, CCF2-AM, by flow cytometry as described by others [60, 61]. Pmacs exposed to IL-4/IL-13 were more susceptible to VLP fusion (Fig 2C), demonstrating that enhanced entry, likely secondary to enhanced binding, is at least in part responsible for the increased infection observed in M2 macrophages. Importantly, VLPs expressing the native VSV glycoprotein exhibited no increase in fusion in IL-4/IL-13 treated pmacs, providing strong evidence that the increased entry in M2a pmacs is specific to the EBOV GP (Fig 2C).

## IL-4/IL-13 treatment stimulates expression of C-type lectin receptors and, to a lesser degree, phosphatidylserine receptors

Enhanced virion binding and entry into IL-4/IL-13-treated cells suggested that one or more cell surface receptors used by filoviruses are upregulated. To evaluate this, the expression of genes encoding many of the known cell surface receptors used by EBOV was assessed by qRT-PCR in untreated or IL-4/IL-13-treated pmacs. The expression of the CLR, macrophage galactose binding lectin (MGL or CLEC10a), was upregulated ~13-fold by IL-4/IL-13 treat-ment (Fig 3A). In addition, two PS receptors, Tyro3 and TIM-4, and the critical endosomal receptor NPC1 were modestly enhanced, but expression of other known filovirus surface receptors was unchanged. Because expression of Tyro3 and TIM-4 were modestly increased in

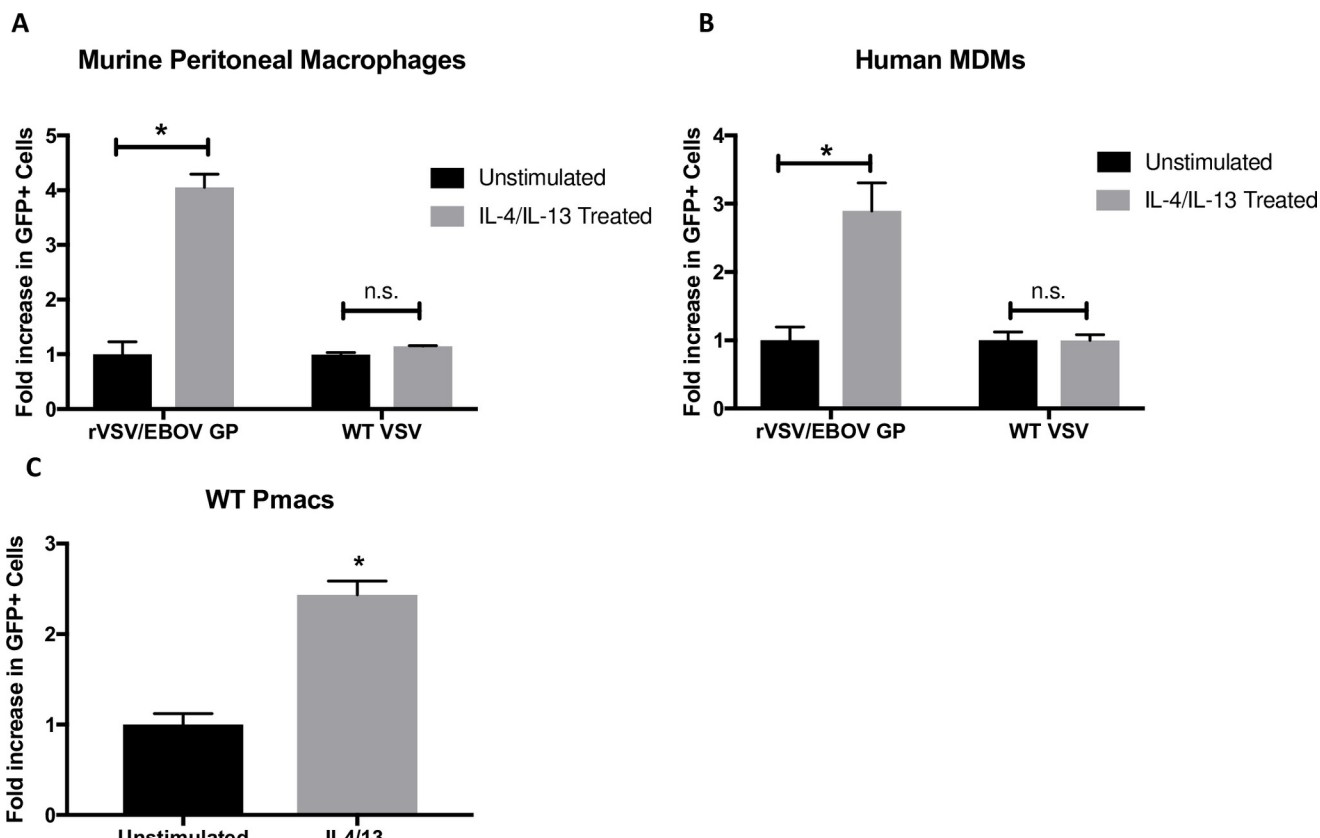

**Fig 1. Macrophage IL-4/IL-13 polarization alters susceptibility to rVSV/EBOV GP infection, but not rVSV/G. A)** Peritoneal macrophages isolated from C57BL/6 *Ifnar*[-/-] mice were treated with 20ng/ml IL-4 and IL-13 or left untreated for 24 hours. Cells were infected with rVSV/EBOV GP(MOI = 0.1) or rVSV/G (MOI = 0.033). As both viruses encode and express GFP, infection was quantified at 24 hours by detection of GFP+ cells by flow cytometry. Shown are the fold increases above unstimulated controls that are set as 1. **B)** *Ex vivo* matured human monocyte derived macrophages were treated with 20ng/ml of IL-4 and IL-13 for 24 hours prior to infection with rVSV/EBOV GP (MOI = 5) or WT VSV (MOI = 1). Infection was quantified at 24 hours by the number of GFP+ cells by flow cytometry and expressed relative to unstimulated controls. **C)** Peritoneal macrophages were harvested from female C57BL/6 mice, treated with 20ng/ml IL-4/IL-13 for 24 hours or left unstimulated, and infected with rVSV/EBOV GP(MOI = 1). Infection was quantified at 24 hours by flow cytometry and expressed relative to unstimulated controls. Data are shown as pooled results from two independent donors representative of many additional replicates. Data are expressed as mean ± S.D. Significance was determined by Student's t-test comparing individual treatments to unstimulated controls, *p<0.05.

the presence of IL-4/IL-13, we incubated IL-4/IL-13 stimulated or unstimulated pmacs in the presence of liposomes composed of phosphatidylserine (PS). These liposomes competitively inhibit the PS-dependent attachment of EBOV to PS receptors, thereby blocking PS-dependent entry [48]. We found that IL-4/IL-13 enhanced virus infection of pmacs by about 3 to 3.5-fold, regardless of whether PS liposomes were present suggesting these receptors do not mediate the IL-4/IL-13-enhanced infection (S2 Fig). We also investigated IL-4/IL-13 upregulation of DC-SIGN on human MDMs as well as the murine orthologs on pmacs. IL-4/IL-13 treatment has been previously shown to upregulate expression of DC-SIGN on human dendritic cells [62–64]. Consistent with these reports, we found that human MDMs expressed significantly higher amounts of DC-SIGN RNA and the number of MDMs expressing DC-SIGN on their cell surface increased when stimulated with IL-4 and IL-13 (S3 Fig). Mice encode 8 potential DC-SIGN orthologs and reports differ on whether SIGNR1, 3 or 5 is most functionally similar to human DC-SIGN [65–67]. We found that IL-4/IL-13 treatment dramatically enhanced SIGNR3 and SIGNR5 RNA expression in pmacs, but not SIGNR1 (Fig 3B), suggesting that SIGNR3 and/or SIGNR5 are similar to DC-SIGN in responsiveness to IL-4/IL-13.

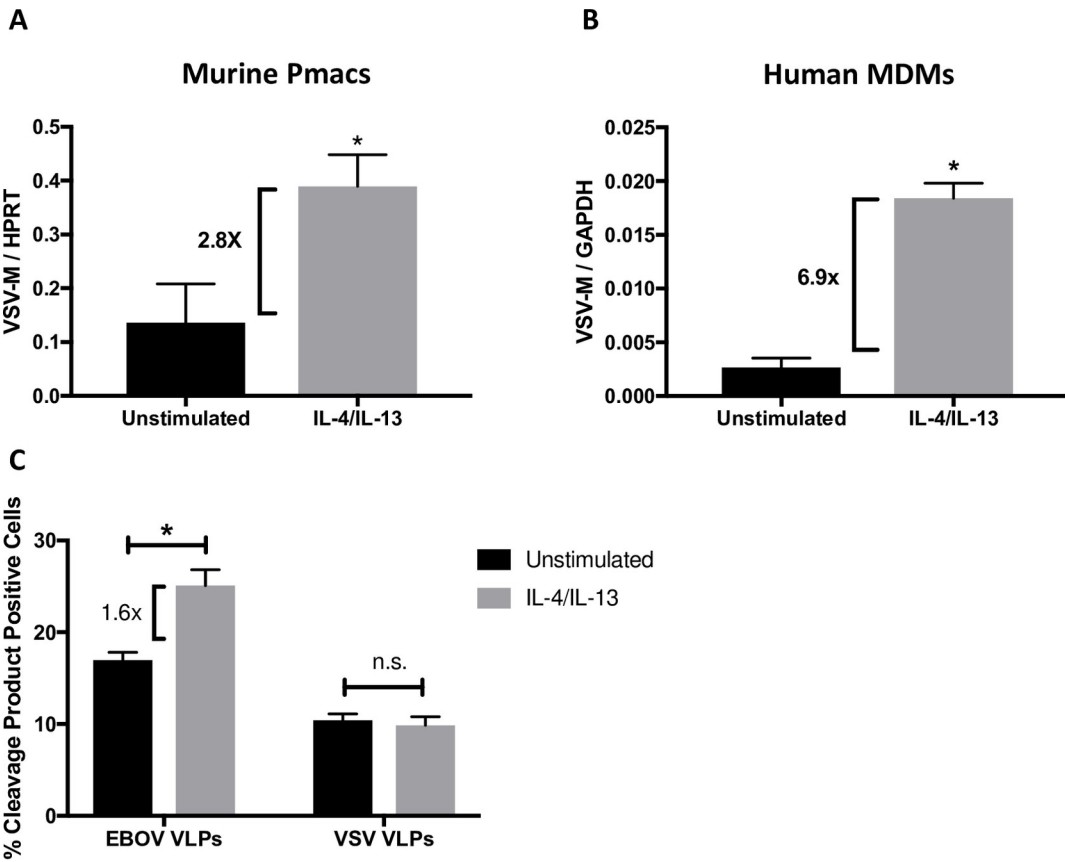

**Fig 2. Early entry events of rVSV/EBOV GP are enhanced in M2a polarized macrophages. A,B)** Virus binding to cells. Murine peritoneal macrophages (**A**) or human monocyte derived macrophages (**B**) were polarized to an M2a phenotype with 20ng/ml IL-4/IL-13 for 24 hours. Polarized cells were placed on ice and incubated at 4°C for 1 hour in the presence of rVSV/ EBOV GP, MOI = 2, to allow for binding but not internalization. Cells were subsequently washed with PBS to remove unbound virus and RNA was isolated for qRT-PCR analysis of viral genomes. Findings are expressed as VSV-M (viral genome) relative to levels of housekeeping genes, HPRT (pmacs) or GAPDH (MDMs). **C)** Virion fusion efficiency. Peritoneal macrophages from *Ifnar*[-/-] mice were incubated with VLPs expressing either EBOV GP or VSV G and EBOV VP40 fused to beta lactamase (detailed in Materials and Methods). Entry was quantified by loading cells with CCF2-AM, a β-lactam-containing fluorescent substrate, and run on flow cytometry to determine the relative amount of cleaved and un-cleaved species. Data are shown as mean ± S.D. Each experiment was performed a minimum of two times. Statistics were calculated using Student's t-test, *p<0.05.

Additionally, the number of cells within the pmac population expressing cell surface CLRs, SIGNR3, SIGNR5, and MGL, were enhanced by the cytokine treatment; however, neither SIGNR1 nor the PS receptor, TIM-4, were increased (Fig 3C). Parallel surface staining studies were unable to detect expression of Tyro3. These data suggested that CLRs may be responsible for enhanced EBOV GP-dependent entry into IL-4/-13 treated pmacs.

To evaluate which murine SIGNR family members are capable of enhancing rVSV/EBOV GP infection, we transfected expression plasmids of DC-SIGN, SIGNR1, 3 and 5 into HEK 293T cells which lack known EBOV surface receptors and are poorly permissive for infection. Transfection of human DC-SIGN, used as a positive control, resulted in a ~35-fold enhancement of rVSV/EBOV GP infection (Fig 3D). Interestingly, transient expression of SIGNR3 and SIGNR1 increased EBOV infection, albeit to lower levels than DC-SIGN, whereas SIGNR5 expression increased infection less than two-fold. Importantly, none of these receptors when expressed in HEK 293T cells impacted rVSV/G infection to the degree seen in rVSV/EBOV GP, although DC-SIGN and SIGNR5 achieved statistical significance (S4 Fig).

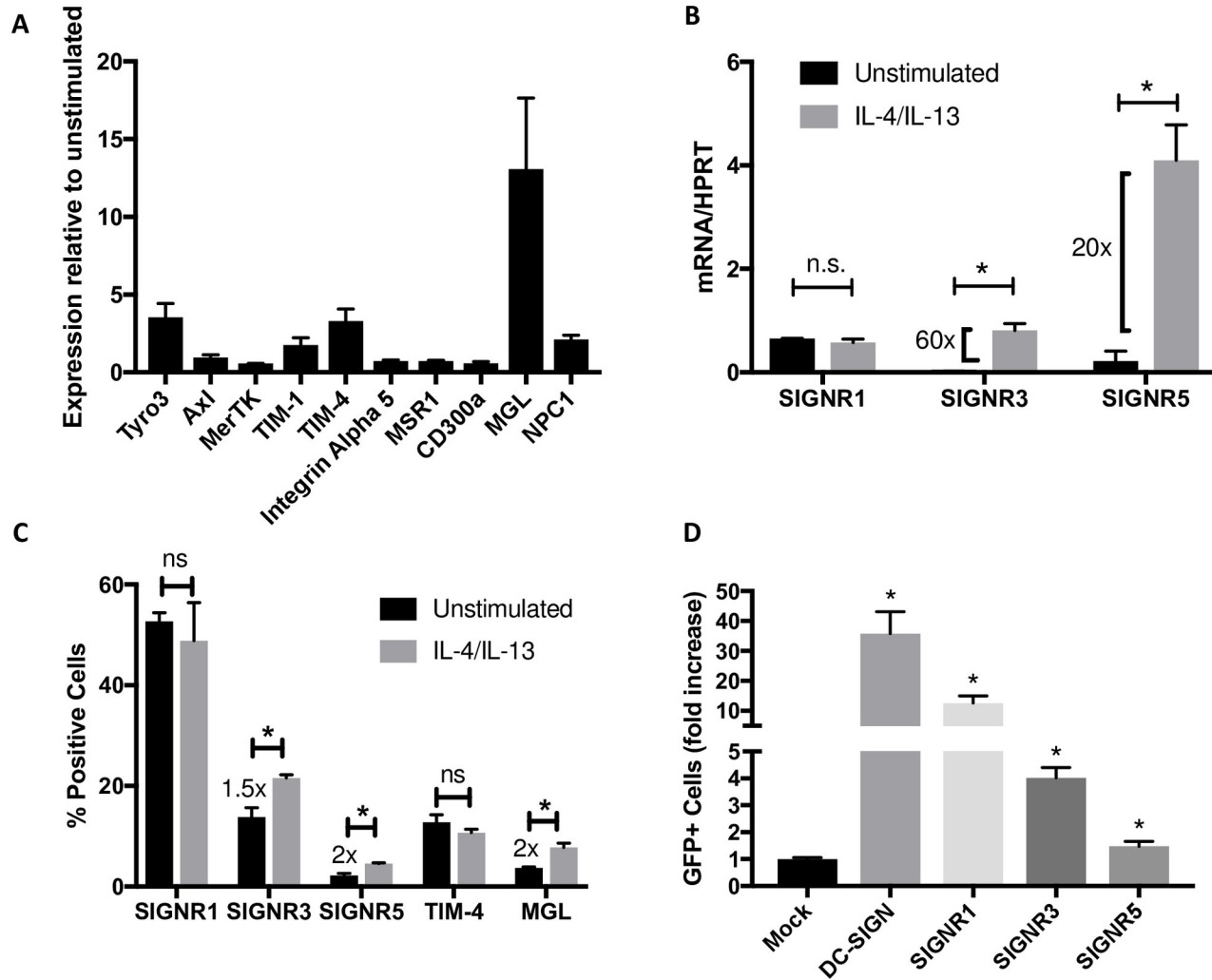

**Fig 3. SIGNR3 expression is elevated in IL-4/IL-13 polarized macrophages. A)** Analysis of expression of known surface receptors for EBOV in C57BL/6 *Ifnar*^-/- pmacs that were unstimulated or stimulated with 20ng/ml of IL-4/IL-13 for 24 hours. Shown is expression relative to unstimulated controls. **B)** Expression of SIGNR genes in unstimulated or IL-4/IL-13 stimulated pmacs. Expression was quantified by qRT-PCR and expressed relative to HPRT levels. **C)** Cell surface expression of EBOV surface receptors in unstimulated or IL-4/IL-13 stimulated cells. Surface expression was evaluated by immunostaining followed by flow cytometry. **D)** HEK 293T cells were transfected with plasmids expressing the indicated proteins and infected with rVSV/EBOV GP (MOI = 1). Infection was quantified by GFP detection at 24 hours post infection by flow cytometry. Data are shown as mean ± S.D. Each experiment was performed 3 times. Statistical analyses were performed using Student's t-test, *p<0.05.

## Mannan, but not GalNac, inhibits IL-4/-13 enhancement of virus infection

With the demonstration that expression of human DC-SIGN, murine SIGNR3 and MGL was increased in IL-4/IL-13-stimulated macrophages, we assessed the effect of these receptors on virus infection using carbohydrate-based competitors. HEK 293T cells were transfected with DC-SIGN, SIGNR3, or MGL and infected with rVSV/EBOV GP in the presence or absence of mannan or N-acetyl galactosamine (GalNac) that bind to DC-SIGN [68] and SIGNR3 or MGL, respectively [69]. Addition of mannan, but not GalNac, reduced infection of DC-SIGN or SIGNR3 transfected cells, whereas GalNac blocked MGL dependent infection (Fig 4A). Mannan, but not GalNac, specifically reduced virus infection of IL-4/IL-13 polarized murine pmacs and human MDMs (Fig 4B and 4C), indicating that mannose binding receptors were responsible for mediating virus uptake. These results, in conjunction with earlier qRT-PCR

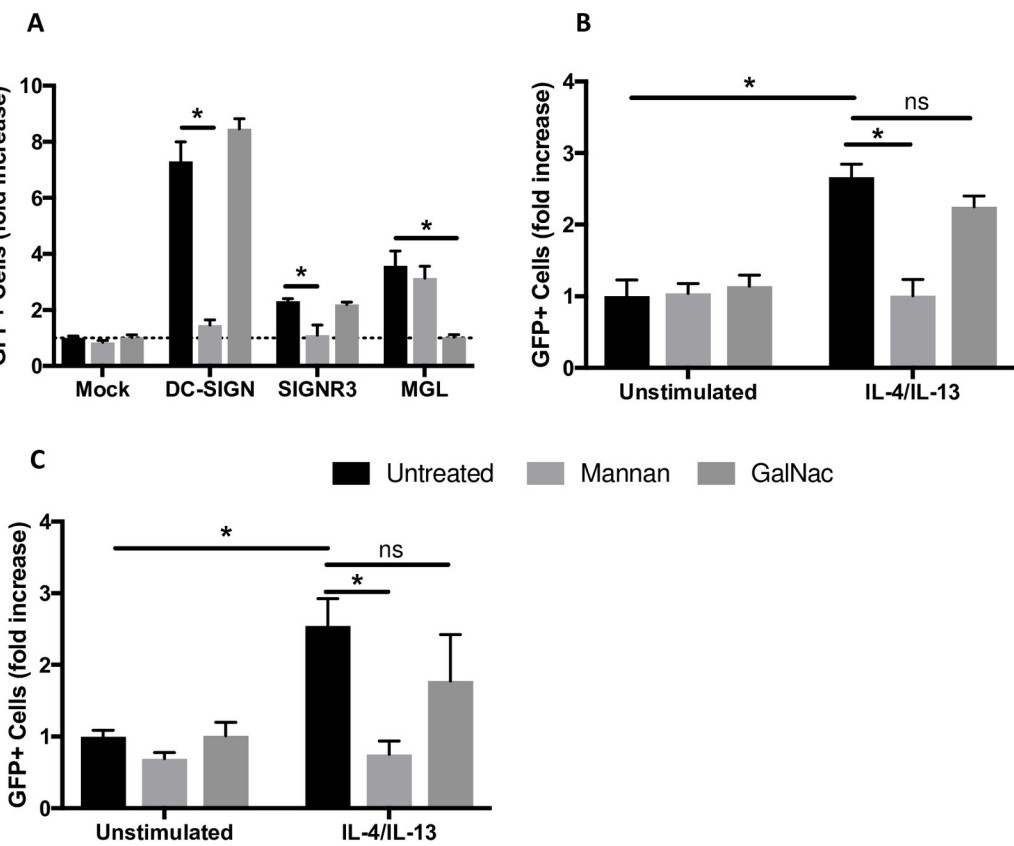

**Fig 4. Competitive inhibition of SIGNR3 and DC-SIGN reduces rVSV/EBOV GP infection. A**) HEK 293T cells were transfected with plasmids expressing the indicated proteins. Forty-eight hours after transfection, cells were treated with the indicated sugar for one hour and infected with rVSV/EBOV GP (MOI = 1). GFP+ cells were quantified 24 hours post infection. **B-C**) Peritoneal macrophages from C57BL/6 *Ifnar*−/− mice (**B**) or human monocyte derived macrophages (**C**) were isolated and polarized with IL-4/IL-13 for 24 hours. Cells were exposed to 200μg/mL of mannan or GalNac for 1 hour prior to infection. Cells were infected with rVSV/EBOV GP, MOI = 0.1 and GFP+ cells were quantified 24 hours post infection. Data are shown as mean ± S.D. Each experiment was performed 3 times. Statistical analysis was performed using Student's t-test, *p<0.05.

data, strongly suggest that DC-SIGN/SIGNR3 upregulation mediates increased rVSV/EBOV GP susceptibility in M2a polarized murine macrophages. Further, our findings indicate that SIGNR3 in murine pmacs acts in a manner most similar to DC-SIGN in hMDMs with regards to EBOV infection.

## Loss of peritoneal macrophage TIM-4 expression enhances the effect of IL-4/IL-13 on rVSV/EBOV GP infection

While the impact of IL-4/IL-13 upregulation of mannose-binding CLRs such as SIGNR3 on virus infection of pmacs is statistically significant, it is a modest effect. We postulated that one possible explanation for the limited enhancement by IL-4/IL-13 of virus infection in pmacs is that other receptors utilized by EBOV are also available and utilized in pmacs. Thus, we investigated the use of another source of primary macrophages, the bone marrow, and found that bone marrow derived, M-CSF matured macrophages (BMDMs) poorly support rVSV/EBOV GP infection. Further IL-4/IL-13 stimulation does not upregulate SIGNR3 expression or enhance infection (S5 Fig). Hence, we returned to the use of pmacs. Murine pmacs not only express SIGNR3 upon IL-4/-13 treatment, but constitutively express the PS receptor TIM-4

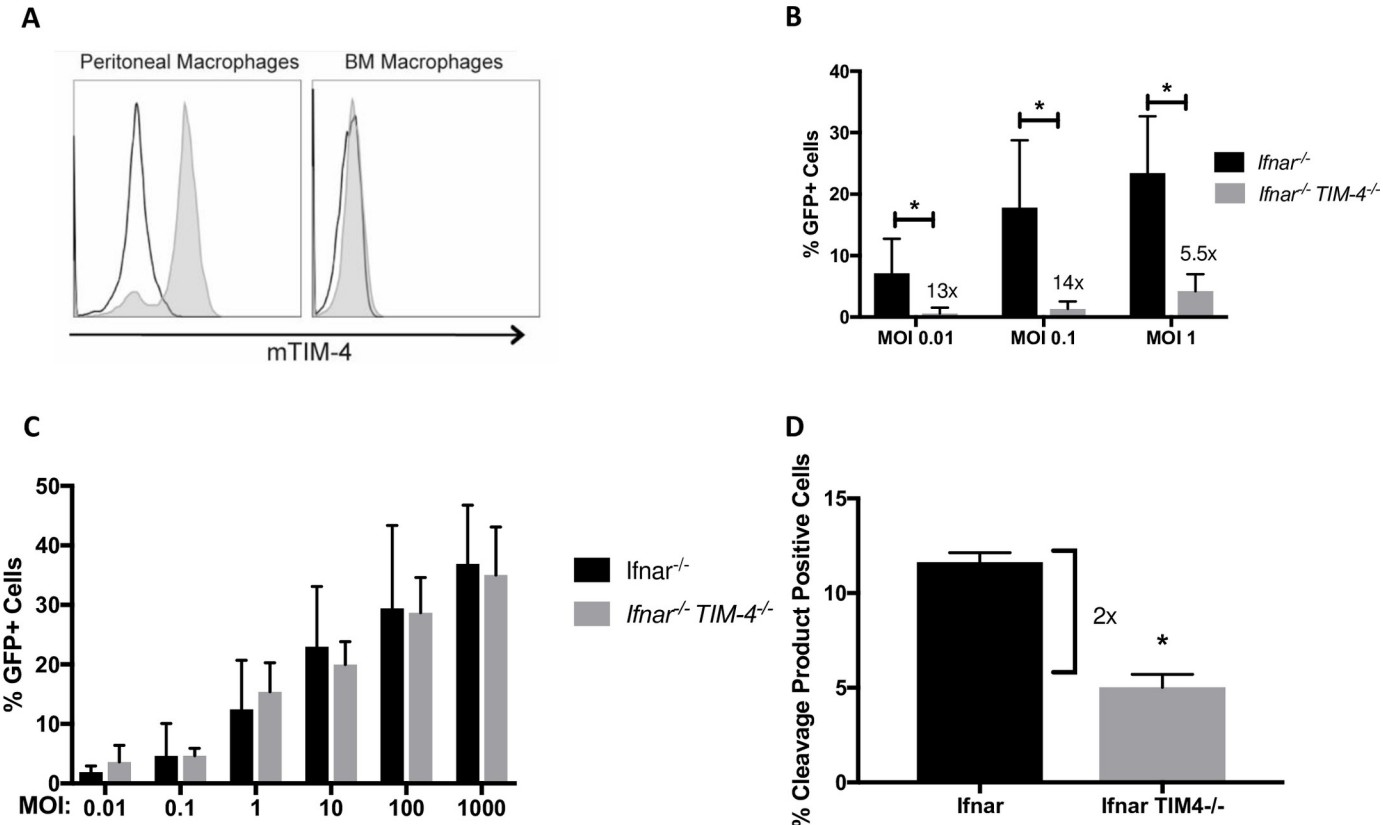

**Fig 5. TIM-4 is a critical EBOV receptor on peritoneal macrophages. A)** TIM-4 surface expression. Matured bone marrow derived macrophages and pmacs from C57BL/6 *Ifnar*[-/-] mice were lifted from tissue culture plates, stained with directly conjugated F4/80 and CD11b mAb. Positively gated cells were analyzed for TIM-4 (grey histogram) or isotype control (white histogram) by flow cytometry. **B)** TIM-4 expression is required for robust rVSV/EBOV GP infection. C57BL/6 *Ifnar*[-/-] or *Ifnar/Timd-4*[-/-] pmacs were infected with the indicated infectious units of rVSV/EBOV GP. Twenty-four hours following infection, cells were quantified for GFP expression by flow cytometry. **C)** Infection of BMDMs is not affected by the absence of TIM-4 expression. Bone marrow cells were isolated from C57BL/6 *Ifnar*[-/-] or *Ifnar/Timd-4*[-/-] mice. Adherent cells were matured into macrophages by incubating for 6 days in the presence of 50ng/mL MCSF. Matured macrophages were infected with the indicated infectious units of rVSV/EBOV GP and number of infected cells were quantified by GFP expression at 24 hour following infection by flow cytometry. **D)** Pmacs were harvested from C57BL/6 *Ifnar*[-/-] or *Ifnar/Timd-4 -/-* mice and incubated with VLPs expressing EBOV GP and EBOV VP40 fused to beta lactamase. Virus/cell membrane fusion was subsequently quantified by loading cells with CCF2-AM, a β-lactam-containing fluorescent substrate, and analyzed on flow cytometry to determine the relative amount of cleaved and un-cleaved substrate. Data are shown as mean ± S.D. Statistics were calculated using Student's t-test, *p<0.05.

[47], that has been shown to serve as a filovirus receptor in HEK 293T cells [51, 70]. We investigated if constitutive, high level TIM-4 expression mediates filovirus entry into pmacs, making SIGNR3-dependent virus uptake redundant and thereby reducing the impact of SIGNR3 expression on filovirus infection. To test this, we evaluated TIM-4 expression and observed strong expression on pmacs, but no expression on BMDMs (Fig 5A). We also observed that pmacs from TIM-4-sufficient mice supported higher levels of viral infection that those from *Timd-4*[-/-] mice, whereas there was no difference in susceptibility of wild-type versus *Timd-4*[-/-] BMDMs (Fig 5B and 5C). VLP fusion events were also significantly decreased in pmacs from *Timd-4*[-/-] mice when compared to TIM-4-sufficient pmacs (Fig 5D), showing the importance of TIM-4 as a receptor for EBOV in pmacs. These observations explain the poor susceptibility of BMDMs previously observed (S5 Fig). However, as TIM-4 expression was not enhanced by IL-4/IL-13 treatment (Fig 3A), this receptor cannot be responsible for the increase in virus infection mediated by IL-4/IL-13.

We reasoned that infection mediated by TIM-4 obscured the subtler effects of IL-4/IL-13 treatment and the effect of IL-4/IL-13 would likely be more profound in TIM-4-null pmacs.

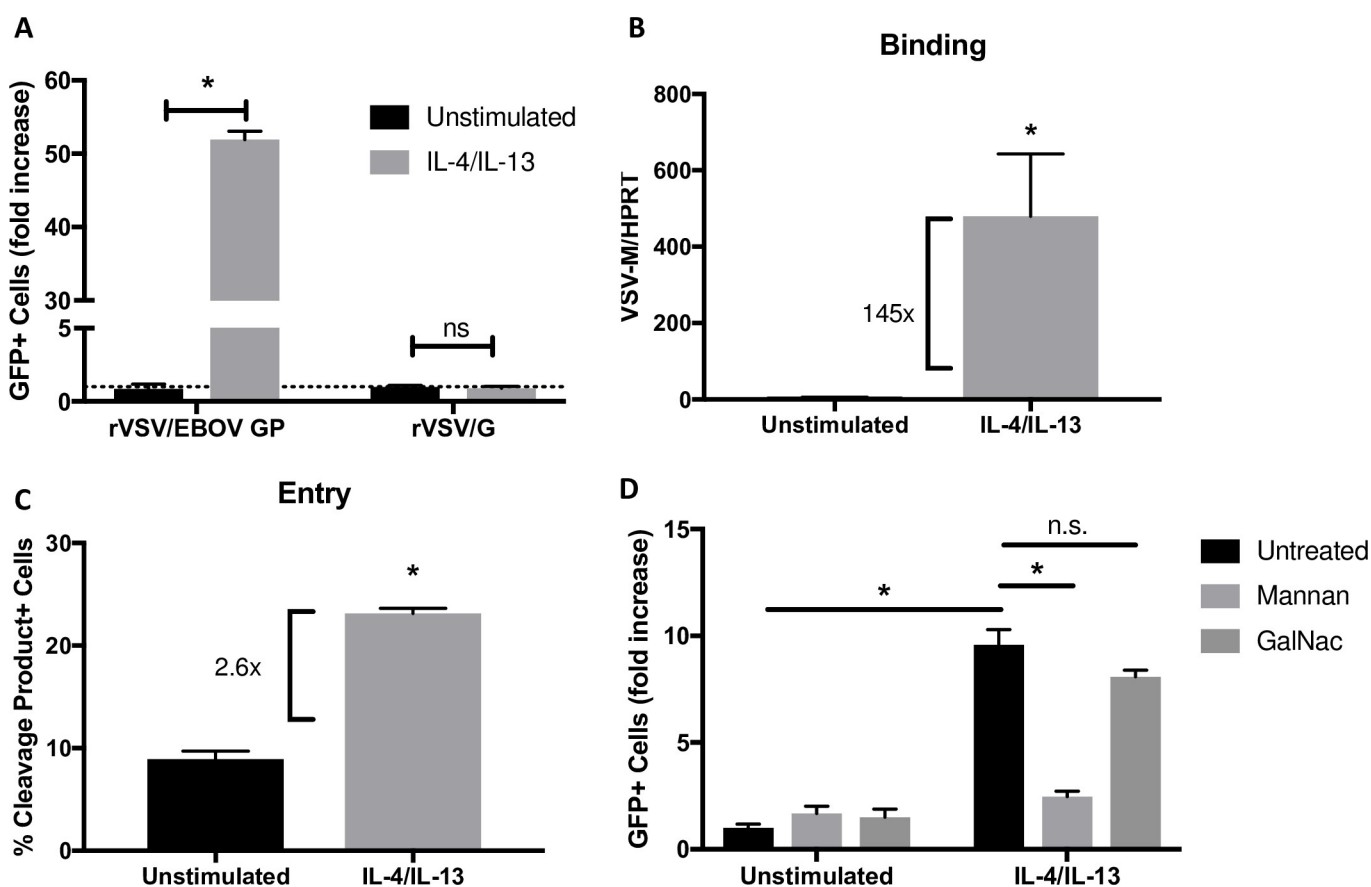

**Fig 6. The effect of IL-4/IL-13 polarization on virus infection is enhanced in TIM-4$^{-/-}$ on peritoneal macrophages.** Pmacs from *Ifnar/Timd-4$^{-/-}$* mice were used in all studies. **A)** IL-4/IL-13 treated or untreated cells were infected with rVSV/EBOV GP (MOI = 0.1) or rVSV/G (MOI = 0.03) and infection was quantified by detection of GFP expression at 24 hours post infection. Expression is shown relative to unstimulated controls. **B)** IL-4/IL-13 treated or untreated cells were placed on ice and infected with rVSV/EBOV GP (MOI = 0.1). After 1 hour, cells were washed with PBS to remove unbound virus and RNA was isolated for qRT-PCR. Results are expressed as VSV-M RNA relative to HPRT control. **C)** IL-4/IL-13 treated or untreated cells were incubated with VLPs expressing EBOV GP and EBOV VP40 fused to β-lactamase. Entry was subsequently quantified by loading cells with CCF2-AM, a β-lactam-containing fluorescent substrate, and analyzed by flow cytometry to determine the relative amount of cleaved and un-cleaved species. **D)** IL-4/IL-13 treated or untreated cells were preincubated with 200ug/mL of mannan or GalNac for 1 hour. Cells were infected with rVSV/EBOV GP (MOI = 0.1) and GFP+ cells were quantified 24 hours post infection. Data are shown as mean ± S.D. Each experiment was performed at least 3 times. Statistical analysis was performed using Student's t-test, *p<0.05.

Indeed, pmacs from *Timd-4$^{-/-}$* mice were more sensitive to the effects of M2a polarization with ~50-fold increase in infection observed in the polarized cells, an effect that was only observed with rVSV/EBOV GP and not with rVSV/G (Fig 6A). Additionally, rVSV/EBOV GP binding and VLP membrane fusion were strongly enhanced in TIM-4-null pmacs by M2a polarization, with a stronger effect on binding as we previously observed (Fig 6B and 6C). Mannan, but not GalNac, inhibited IL-4/IL-13-dependent enhancement in pmacs, providing evidence that a mannose binding CLR such as SIGNR3 was responsible for the cytokine-dependent increase in infection (Fig 6D).

## *In Vivo* M2a Polarization of Murine Pmacs Increases Mortality

To assess the role of macrophage polarization on rVSV/EBOV infection *in vivo*, we challenged control (*Ifnar$^{-/-}$*) and *Ifnar/Timd-4$^{-/-}$* C57BL/6 mice with virus. However, initially, we needed to determine if *in vivo* IL-4/IL-13 treatment would polarize pmacs. We administered the M2a

polarizing stimulus IL-4/IL-13 i.p. and, in parallel, the M1 polarizing stimulus IFN-γ was also tested for its effect. After 24 hours, peritoneal cells (consisting primarily of macrophages, but also containing epithelial cells and other cell types) were isolated and changes in gene expression by qRT-PCR were assessed. We found that expression of the M1 marker gene IRF-1 was modestly increased in mice administered IFN-γ, and that the M2a marker genes Fizz-1 and, in particular, SIGNR3 were increased in IL-4/IL-13 treated mice (S6 Fig). These data suggest that acute *in vivo* polarization of resident peritoneal macrophages occurs following administration of a single dose of polarizing stimuli and provide validation for *in vivo* cytokine polarization studies.

To determine if IL-4/IL-13 enhances rVSV/EBOV GP infection *in vivo*, $Ifnar^{-/-}$ mice were administered either PBS or 10 μg of IL-4/IL-13 i.p. and 24 h later infected i.p. with a sublethal dose of rVSV/EBOV. The mice were monitored daily for survival. We found that *in vivo* M2a polarization of the TIM-4-sufficient mice elicited a modestly enhanced mortality (Fig 7A), whereas rVSV/G challenge was equally virulent to PBS or IL-4/-13 treated mice (Fig 7B). To evaluate the role of peritoneal macrophages in this *in vivo* phenotype, pmacs were harvested from mice 2 days post infection and viral load was assessed by qRT-PCR. We found that IL-4/IL-13 enhanced *in vivo* infection of pmacs upon rVSV/EBOV GP challenge, but not rVSV/G (Fig 7C). These data suggested that *in vivo* IL-4/-13 administration increases virus load in an EBOV GP-dependent manner. In parallel, $Ifnar/Timd-4^{-/-}$ mice were challenged as we anticipated that these mice would be even more sensitive to the effect of IL-4/IL-13 treatment. In these studies, ~100-fold higher concentrations of rVSV/EBOV GP were administered since we observed that TIM-4-null mice were decidedly less permissive for the virus and we observed that loss of TIM-4 protected mice from virus challenge (Fig 7D). In $Ifnar/Timd-4^{-/-}$ mice, a dose of rVSV/EBOV GP that resulted in no mortality in the PBS-treated mice caused 100% mortality in IL-4/IL-13-treated mice (Fig 7E). Because we recognize that *in vivo* administration of a large bolus of cytokine is likely to have additional effects on the immune response, we performed *ex vivo* polarization and adoptive transfer of IL-4/IL-13 treated macrophages prior to challenge of $Ifnar/Timd-4^{-/-}$ mice with rVSV/EBOV GP. As observed with *in vivo* polarization, these mice had increased mortality (Fig 7F) compared to mice receiving PBS-treated macrophages, although the adoptive transfer may have increased the sensitivity of the mice to virus since more than half of the PBS-treated mice also succumbed to virus infection under these conditions. Taken together these data provide evidence that M2a polarization exacerbates EBOV infection *in vivo* likely due to the enhanced infection of a critical cell population, peritoneal macrophages.

## Discussion

We demonstrate in these studies that IL-4/IL-13 polarization increases the expression of cell surface receptors important for EBOV infection on mouse pmacs and human MDMs, resulting in enhanced Ebola GP-dependent infection in both wild-type and $Ifnar^{-/-}$ cells. *In vivo* administration of IL-4/IL-13 or addition of IL-4/IL-13-treated pmacs to $Ifnar/Timd-4^{-/-}$ mice resulted in greater viral pathogenesis as reflected in increased mortality. We further demonstrate both *in vivo* and *in vitro* that IL-4/IL-13 enhances EBOV GP specific entry, as rVSV/G infection of mice and pmacs were not altered by these cytokines. We propose that our findings extend to WT EBOV and potentially other filoviruses as the mechanism of entry is highly conserved.

It is important to highlight the use of a model virus in these experiments. rVSV/EBOV GP has been found to be an excellent model for understanding EBOV entry, recapitulating all steps of binding and internalization. However, it is less well-suited to study filovirus

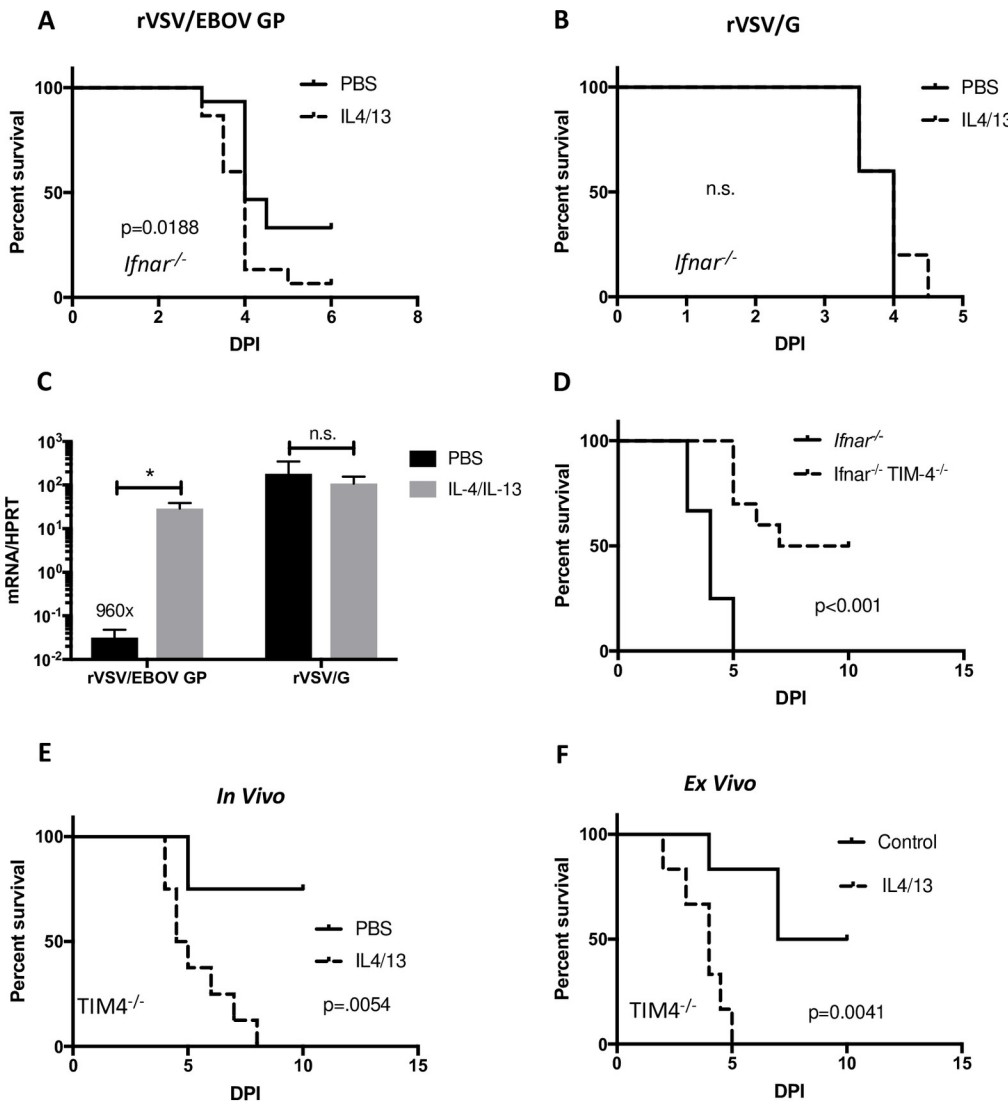

**Fig 7. Macrophage polarization increases morbidity and mortality in a mouse model. A-C)** Female *Ifnar*[-/-] mice were injected i.p. with 10ug IL-4/IL-13 or PBS 24 hours prior to i.p. infection with either 7x10$^1$ iu rVSV/EBOV GP (n = 10) or 1x10$^2$ iu rVSV/G (n = 5). Survival was monitored **(A,B)** or pmacs were isolated at day 2 post infection and assessed for viral load by qRT-PCR **(C)**. **D)** Female *Ifnar*[-/-] (n = 12) or *Ifnar/Timd-4*[-/-] (n = 10) mice were infected i.p. with 10$^4$ iu of rVSV/EBOV GP and monitored daily for mortality. **E)**. Male *Ifnar/Timd-4*[-/-] mice were injected i.p. with 10ug IL-4/IL-13 or PBS 24 hours prior to i.p. infection with 1x10$^4$ i.u. rVSV/EBOV GP. Survival was monitored (n = 8/ group). **F)** Peritoneal macrophages from male *Ifnar/Timd-4*[-/-] mice were isolated and polarized to an M2a phenotype or treated with MCSF (controls) for 24 hours. Polarized cells were lifted and injected i.p. into recipient male *Ifnar/Timd-4*[-/-] mice (1.5x10$^6$ cells/mouse). Twenty-four hours after transfer of macrophages, mice were infected i.p. with 1x10$^4$ i.u. rVSV/EBOV GP and survival was monitored (n = 6/group). All in vivo experiments were performed a minimum of two times. Statistical analyses for all in vivo graphs were performed by Mantel-Cox test, p values indicated. For figure C statistics were performed by Student's T-test, * indicates p<0.05.

pathogenesis as post-entry events are dictated by the VSV genome. Thus, our *in vivo* studies included rVSV/G controls that enabled us to demonstrate that IL-4/IL-13 enhanced pathogenesis *in vivo* was specific for virus encoding EBOV GP. Future BSL-4 experiments with EBOV would be valuable in extending our findings.

It should also be noted that our model virus necessitated the use of *Ifnar*[-/-] mice in our survival studies. Further, many of our tissue culture studies were performed in *Ifnar*[-/-] pmacs,

although WT pmacs and human MDMs were evaluated as well and found to respond to IL-4/IL-13 in a similar manner as the *Ifnar*<sup>-/-</sup> cells. In addition to a lack of type I IFN signaling, *Ifnar*<sup>-/-</sup> cells have modestly altered adaptive immune responses [71, 72]. Thus, the use of these mice is potential caveat to our studies.

We show that M2a polarization of pmacs enhances virus replication and pathogenesis through upregulation of expression of CLRs on the cell surface. Mannose-binding CLRs, DC-SIGN in humans and SIGNR3 in mice, were increased on the surface of pmacs by a 24 h treatment with IL-4/IL-13. Expression of MGL, another CLR that can support EBOV binding and internalization [11, 73], was also significantly upregulated by IL-4/IL-13 treatment. However, while mannose polymer, mannan, abrogated IL-4/IL-13 enhancement, the MGL ligand, GalNAc, did not, despite a similar level of receptor upregulation. This suggests that DC-SIGN/SIGNR3 are more efficient receptors on macrophages, but does not inherently disagree with other reports that MGL can mediate entry in other contexts. We also note that there were changes in other potential receptors, including the endosomal receptor NPC1, that could contribute to the enhanced entry upon IL-4/IL-13 exposure, however studies with mannan suggest that mannose-binding CLRs are critical for this effect.

DC-SIGN has previously been shown to be a receptor for attachment and internalization of EBOV and not a receptor for VSV [8, 74]. However, attempts to identify mouse orthologs with functionality similar to DC-SIGN have met with limited success. Baribaud, et al (2001) cloned a murine cDNA that was initially proposed to be a functional murine ortholog [75]. This cDNA was shown to have 68% identity to human DC-SIGN and bound to ICAM-3 and HIV-1 and HIV-2 in a manner similar to human DC-SIGN [75]. However, the authors noted that expression of this cDNA did not result in efficient transmission of HIV to T cells in a mixed culture, suggesting this receptor did not entirely recapitulate the behavior of human DC-SIGN [75]. This murine cDNA was shown by others to encode murine SIGNR1 [66]. The murine CLR, CIRE or SIGNR5, was also evaluated as the murine homolog to human DC-SIGN [76]. However, SIGNR5 was not found to bind to EBOV GP and the authors concluded that mice do not have a DC-SIGN homolog that served as a cell surface receptor for EBOV. We find that both SIGNR1 (CD209b) and SIGNR3 (CD209d) are able to mediate virus entry, whereas SIGNR5 (CD209a) was a less effective receptor. These findings are consistent with the work by Gramberg et al [76] as well as previous studies that found that SIGNR1 and SIGNR3 bind mannose and complex mannose containing glycans in a manner similar to DC-SIGN and possess endocytic capabilities [77], whereas SIGNR5 lacks that crucial endocytic function [66]. While both SIGNR1 and SIGNR3 mediated rVSV/EBOV GP infection, only SIGNR3 is upregulated when exposed to IL-4/IL-13. As additional evidence, we also show that IL-4/IL-13 exposure does not increase SIGNR3 expression in BMDMs and these polarized macrophages do not exhibit IL-4/-13-dependent enhancement of rVSV/EBOV GP infection. In total, these data provide strong evidence that SIGNR3 upregulation is a key component of the IL-4/IL-13-mediated increase in virus infection in murine macrophages. Thus, we propose that murine SIGNR3 functions in a similar manner to human DC-SIGN in the context of EBOV infection during M2a polarization. This role for SIGNR3 has not been previously recognized.

On a structural basis, it is somewhat unexpected that murine SIGNR1 or SIGNR3 mediate EBOV binding and internalization. A 191 amino acid extracellular neck domain of DC-SIGN composed of 7 repeats is thought to be important for oligomerization of the receptor [78–80], facilitating high affinity binding of its carbohydrate ligand. A similar long repeat domain is absent in the murine proteins; the neck domains of SIGNR1 and SIGNR3 are shorter, containing ~100 and ~68 residues, respectively. However, apparently the neck domain of SIGNR3 is sufficient for homo-oligomerization, as it has been shown to weakly form dimers upon ligand interaction [65]. An added structural issue with these CLRs is that an 8 amino acid motif

composing the DC-SIGN carbohydrate binding domain is not completely conserved [81–83], potentially resulting in reduced glycan binding.

This study provides evidence for the importance of macrophages in EBOV infections. We have previously shown that IFN-γ-treated macrophages are profoundly resistant to EBOV infection and that IFN-γ-treated mice are protected against EBOV [33]. Further, adoptive transfer of M1 murine pmacs protects mice challenged i.p. with an otherwise lethal dose of rVSV/EBOV GP (Rogers, et al, manuscript under review). Here, we show that "M2" macrophages exert an opposite effect, with this polarization exacerbating rVSV/EBOV GP infection both *in vitro* and *in vivo*. These findings indicate that pmacs play a pivotal role in rVSV/EBOV GP pathogenesis, and most likely EBOV pathogenesis, when virus is administered i.p., exerting an effect that far exceeds their prevalence in the body. Future studies are needed to investigate the role of tissue macrophages when virus is administered via other routes.

It is of interest to understand physiological or pathological conditions that may push macrophage populations to an "M2" state and how this might influence the course of filovirus infection. Disease states eliciting a strong Th2 response may skew macrophages to an "M2" phenotype, although many chronic diseases are complex and appreciated to elicit a mixed M1/M2 phenotype under some circumstances [21]. Two well established disease states that elicit an M2-like phenotype are parasitic infections caused by roundworm and whipworms and tumors [84]. Worms elicit a strong type 2 immune response and may play an important role as a pre-existing condition during filovirus outbreaks. It is estimated that over 1 billion individuals worldwide are infected with parasitic worms, with the greatest prevalence in Sub-Saharan Africa where EBOV is co-endemic [85]. Furthermore, the WHO estimates that over 800 million children are in need of treatment from these pathogens, highlighting the prevalence of infection and difficulty in providing care. Given our findings that M2a cytokines elicit enhanced virus infection in macrophages, future studies are warranted to understand the role of helminth infections during EBOV co-infection as there may be profound clinical implications. Macrophages present in tumors (TAMs) also express an anti-inflammatory phenotype that is thought to help drive tumorigenesis [84, 86]. While no studies have investigated whether malignancies drive more robust filovirus replication, the loss of immune control combined with the enhanced presence of M2 polarized macrophages found during tumorigenesis would be anticipated to establish an *in vivo* setting conducive for filovirus infection.

Of note, EBOV infection itself may alter the macrophage polarization state. EBOV GP interacts with and signals through TLR4 [87–89]. However, the effect of this interaction on macrophage polarization has not been directly examined. While LPS/TLR4 interactions can generate a M1 polarization state [22], in combination with immune complexes LPS stimulates a M2b state [90]. Thus, it would be of interest to examine the effect of EBOV GP on macrophage gene expression.

Finally, our studies also highlight the importance of TIM-4 for EBOV GP-dependent entry into murine pmacs. We have previously shown that ectopic expression of human and mouse TIM-4 in HEK 293T cells enhances rVSV/EBOV GP entry [50, 51]. While it is appreciated that tissue macrophage and dendritic cell populations express endogenous TIM-4 in both humans and mice [91, 92], the role of this receptor in filovirus infection of those primary populations has not been previously reported. Two lines of evidence indicated that TIM-4 is important for EBOV infection of pmacs. First, infection of TIM-4-sufficient pmacs is inhibited by addition of PS liposomes and, second, virus infection of pmacs from *Timd-4*<sup>-/-</sup> is markedly lower inTIM-4-sufficient pmacs. Consistent with an important role for TIM-4 as a receptor in pmacs that can mask the effect of IL-4/IL-13 polarization, the effect of IL-4/IL-13 was notably enhanced when TIM-4 was absent. In total, our studies have been able to tease apart the multiple layers of host receptor-dependent events that drive filovirus entry and infection.

## Supporting information

**S1 Fig. Gene expression profile of polarized macrophages.** Murine peritoneal macrophages harvested from C57BL/6 Ifnar-/- mice were plated in media containing 50 ng/mL M-CSF for 48 hours. Media was refreshed with some wells incubated in media with 20 ng/ml IL-4 and IL-13 for 24 hours. RNA was isolated and qRT-PCR was performed to analyze expression of genes known to be characteristic of polarized macrophages (n = 5 per treatment). Gene expression is shown relative to the housekeeper gene HPRT. Data are shown as mean ± S.D. and represent at least three independent replicates. Significance was determined by Student's t-test comparing individual treatments to unstimulated controls, *p<0.05.
(TIF)

**S2 Fig. PS receptors do not mediate IL-4/IL-13-dependent increases in rVSV/EBOV GP infection.** Pmacs from *Ifnar*$^{-/-}$ mice were polarized with IL-4/IL-13 or left unpolarized. Twenty-four hours after polarization, cells were incubated for one hour in the presence of liposomes containing phosphatidylserine or plain media. Cells were infected with rVSV/EBOV GP and quantified for GFP expression by flow cytometry after 24 hours. Data are shown relative to levels of infection in pmacs not stimulated with IL-4/-13. Statistics were performed with Student's t-test, * indicates p value < 0.05.
(TIF)

**S3 Fig. DC-SIGN expression in human monocyte derived macrophages.** Human monocyte derived macrophages were polarized with 20 ng/ml IL-4/IL-13 for 24 hours. Levels of DC-SIGN were detected by qRT-PCR (A) and surface staining (B). Statistics were performed with Student's t-test, * indicates p value < 0.05
(TIF)

**S4 Fig. rVSV/G does not utilize SIGNR receptors for cellular entry.** HEK 293T cells were transfected with plasmids expressing the indicated proteins and infected with rVSV/G (MOI = 1). Cells were analyzed 24 hours following infection for GFP expression by flow cytometry. Data are shown as mean ± S.D. Experiment was performed 2 times. Statistics were performed with Student's t-test, * indicates p value < 0.05
(TIF)

**S5 Fig. Infection and polarization of murine bone marrow derived macrophages.** Matured bone marrow derived macrophages from *Ifnar*-/- mice were polarized with 20 ng/mL of IL-4/IL-13 for 24 hours. RNA was harvested and gene expression was analyzed (A) or cells were infected with rVSV/EBOV GP and infection was quantified for GFP expression by flow cytometry at 24hpi (B). Experiment was performed 2 times. Statistics were performed with Student's t-test, * indicates p value < 0.05
(TIF)

**S6 Fig. In vivo polarization.** C57BL/6 *Ifnar*-/- mice were injected with 10 μg IL-4/IL-13. After 24 hours, peritoneal cells were harvested and RNA was isolated for qRT-PCR. Markers associated with macrophage polarization were assessed. Data are expressed as a delta delta Ct (Gene/HPRT relative to Control)
(TIF)

## Acknowledgments

The authors would like to thank Drs. Noah Butler, Al Klingelhutz and Richard Roller for their thoughtful comments on this manuscript.

## Author Contributions

**Conceptualization:** Bethany Brunton, Wendy Maury.

**Data curation:** Kai J. Rogers, Laura Mallinger, Dana Bohan, Wendy Maury.

**Formal analysis:** Kai J. Rogers, Bethany Brunton, Wendy Maury.

**Funding acquisition:** Wendy Maury.

**Investigation:** Kai J. Rogers, Bethany Brunton, Laura Mallinger, Dana Bohan, Kristina M. Sevcik, Jing Chen, Natalie Ruggio.

**Methodology:** Kai J. Rogers, Bethany Brunton, Natalie Ruggio, Wendy Maury.

**Project administration:** Wendy Maury.

**Resources:** Wendy Maury.

**Supervision:** Wendy Maury.

**Validation:** Kai J. Rogers.

**Visualization:** Kai J. Rogers, Wendy Maury.

**Writing – original draft:** Kai J. Rogers.

**Writing – review & editing:** Bethany Brunton, Wendy Maury.

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
