## [Decision Letter · Decision Letter 0]

15 Sep 2019

Dear Dr. Maury:

Thank you very much for submitting your manuscript "IL-4/IL-13 polarization of macrophages enhances Ebola virus glycoprotein-dependent infection" (PNTD-D-19-01182) for review by PLOS Neglected Tropical Diseases. Your manuscript was fully evaluated at the editorial level and by independent peer reviewers. The reviewers appreciated the attention to an important topic but identified some aspects of the manuscript that should be improved.

We therefore ask you to modify the manuscript according to the review recommendations before we can consider your manuscript for acceptance. Your revisions should address the specific points made by each reviewer.

(1) A letter containing a detailed list of your responses to the review comments and a description of the changes you have made in the manuscript.

(2) Two versions of the manuscript: one with either highlights or tracked changes denoting where the text has been changed (uploaded as a "Revised Article with Changes Highlighted" file ); the other a clean version (uploaded as the article file).

(3) If available, a striking still image (a new image if one is available or an existing one from within your manuscript). If your manuscript is accepted for publication, this image may be featured on our website. Images should ideally be high resolution, eye-catching, single panel images; where one is available, please use 'add file' at the time of resubmission and select 'striking image' as the file type. 

Please provide a short caption, including credits, uploaded as a separate "Other" file. If your image is from someone other than yourself, please ensure that the artist has read and agreed to the terms and conditions of the Creative Commons Attribution License at http://journals.plos.org/plosntds/s/content-license (NOTE: we cannot publish copyrighted images). 

(4) Appropriate Figure Files 

Please remove all name and figure # text from your figure files upon submitting your revision. Please also take this time to check that your figures are of high resolution, which will improve both the editorial review process and help expedite your manuscript's publication should it be accepted. Please note that figures must have been originally created at 300dpi or higher. Do not manually increase the resolution of your files. For instructions on how to properly obtain high quality images, please review our Figure Guidelines, with examples at: http://journals.plos.org/plosntds/s/figures

While revising your submission, please upload your figure files to the Preflight Analysis and Conversion Engine (PACE) digital diagnostic tool, https://pacev2.apexcovantage.com/ PACE helps ensure that figures meet PLOS requirements. To use PACE, you must first register as a user. Then, login and navigate to the UPLOAD tab, where you will find detailed instructions on how to use the tool. If you encounter any issues or have any questions when using PACE, please email us at figures@plos.org.

We hope to receive your revised manuscript by Nov 14 2019 11:59PM. If you anticipate any delay in its return, we ask that you let us know the expected resubmission date by replying to this email.

To submit your revised files, please log in to https://www.editorialmanager.com/pntd/

Sincerely,

Thomas Geisbert

Associate Editor

Scott Weaver

Deputy Editor

Reviewer's Responses to Questions

**Key Review Criteria Required for Acceptance?**

**Methods**

-Are the objectives of the study clearly articulated with a clear testable hypothesis stated?

-Is the study design appropriate to address the stated objectives?

-Is the population clearly described and appropriate for the hypothesis being tested?

-Is the sample size sufficient to ensure adequate power to address the hypothesis being tested?

-Were correct statistical analysis used to support conclusions?

-Are there concerns about ethical or regulatory requirements being met?

Reviewer #1: All OK; one instance where additional sttistics are needed is noted in Specific Comments

Reviewer #2: Lines 216-220: The authors state that “cultures were incubated for 24 h in the plating media without MCSF with: 20ng/mL IL-4 + 20ng/mL IL-13 (M2a), 150μg/mL immune complexes (10:1 ratio of anti-ova antibody/ova) and 10ng/mL LPS (M2b), 100nM Dexamethasone (M2c), or 20ng/mL interferon gamma (M1) for 24 hours”. Oddly, the M1, M2b, or M2c data is not presented in the manuscript. A direct infectivity comparison among these macrophage subsets would have made their findings more substantial. This is especially important considering the literature suggests EBOV elicits an LPS-like response via TLR4 activation (M2b-like response), and this mechanism is thought to be GP-dependent. IL-10 is also secreted during EBOV infection, which could facilitate M2c differentiation. 

The peritoneal macrophages used in these studies were selected by adherence rather than FACS or magnetic sorting. This inferior method resulted in 85% purity. Thus, other cells could account for some of the observed effects even though they represent a smaller proportion of total cells.

**Results**

-Does the analysis presented match the analysis plan?

-Are the results clearly and completely presented?

-Are the figures (Tables, Images) of sufficient quality for clarity?

Reviewer #1: Yes, yes, yes

Reviewer #2: The results and figures were clearly presented and of sufficient quality. However, small groups (n=3) for some of the in vivo experiments may have skewed results. For example, Figure 7C and Figure 7D show ifnar/timd4-null mice given the same dose (1000 i.u.) of rVSV/EBOV GP exhibited 50% and 100% survival, respectively. 

Figure 7F states that a “representative” experiment was performed two independent times. Did this data match the other experiment exactly?

**Conclusions**

-Are the conclusions supported by the data presented?

-Are the limitations of analysis clearly described?

-Do the authors discuss how these data can be helpful to advance our understanding of the topic under study?

-Is public health relevance addressed?

Reviewer #1: yes, yes, yes, yes

Reviewer #2: The use of a surrogate virus (recombinant rVSV/EBOV GP) instead of wtEBOV is not ideal but appropriate for these studies considering the advantages of this model (i.e. safety, convenience, record for filovirus entry studies). However, the authors should discuss some of the limitations of this model and how the use of IFNAR deficient cells could skew their results.

No reference is provided showing M2a-specific polarization during helminth infection to justify the focus on this subset (other than IL-4 secretion during EBOV infection). Other cytokines, such as IL-10, are also secreted during EBOV infection, which could polarize macrophages toward the M2c subset. Moreover, EBOV patients co-infected with malaria are 20% more likely to survive, despite this parasite being associated with an M2-skewed phenotype. 

No explanation is provided for the lack of productive rVSV/EBOV GP infection in murine BMDM.

**Editorial and Data Presentation Modifications?**

Reviewer #1: Minor Revision as per Minor ('specific') comments

Reviewer #2: Minor revision. Please see above for specific comments.

**Summary and General Comments**

Reviewer #1: Macrophages are key targets for ebola virus infection, but whether particular types of macrophages are more prone to ebola infection is not known. Here the authors show that treatment with specific cytokines (IL4/IL13) increases expression of DC-SIGN on human macrophages and SIGNR on mouse peritoneal macrophages (macs), with resultant increases in infection by recombinant VSV particles bearing the ebola glycoprotein (GP), but not by particles bearing the VSV G protein. Carbohydrate competition experiments support the contention that the IL4/13-induced increase in macrophage infection is due to mannose receptors, e.g. DCSIGN in human and SIGNR3 in mouse macs. Importantly, they also show that mice given IL4/IL13 or transplanted with macrophages pre-treated with IL4/IL13 show increased susceptibility to rVSV-Ebola GP infection via the IP route. These findings suggest that macrophage polarization, to the M2a subtype, enhances ebola infection and pathogenesis, at least as acknowledged in the IP rVSV challenge model in interferon receptor null mice.

Specific Comments

1. It is suggested to include Supp. Fig 2 (effect IL4/13 treatment on VSV-EBOV GP infection of pMacs from WT mice) as a panel in Fig. 1

2. Line 359: suggested rewrite: “…enhances EBOV GP-mediated binding and entry”

3. Line 371: suggest rewriting: Given the (stronger) enhancement of binding, one can’t say that there is also enhanced fusion per se. The increased entry signal could be because more VLPs are bound.

4. Fig. 2C (and other key Figs): Add the fold-increase notation (as in Fig 2A,B). 

5. Line 388-389: RE: “…suggests…PS receptors in pMacs are important for constitutive virus entry”: either delete or explain better. The statement re: constitutive virus entry does not seem to be supported by the Fig. (Supp. Fig. 3); although it is supported by the literature and subsequent Figures in this manuscript.

6. Supp. Fig. 5 and Lines 409-410: provide statistics. (Are the ~1.5X upticks in GFP+ cells upon transfection of DC-SIGN or SIGNR5 statistically significant?)

7. Line 452: Rephrase: The effect on VLP entry is small relative to effect on VLP binding.

Reviewer #2: Rogers et al. describe the cytokine milieu that promotes EBOV entry. IL-4 and IL-13 polarizes macrophages to an M2a phenotype, which enhances uptake of pseudotyped rVSV-EBOV-GP into murine peritoneal and human monocyte-derived macrophages. The proposed mechanism for this increased infectivity is increased expression of DC-SIGN (or DC-SIGN-like) molecules, which facilitate GP-mediated attachment and entry. Importantly, this work describes murine orthologs that may represent human DC-SIGN equivalents, a controversial topic in the immunology field that needs further clarification. In vivo co-administration of IL-4 and IL-13 increased mortality and enhanced pathogenesis in mice, illustrating the importance of M2a-skewing in the context of EBOV infection. The manuscript is well-written and the experimental design is sound. Moreover, the hypothesis is clearly stated and testable.

PLOS authors have the option to publish the peer review history of their article (what does this mean?). If published, this will include your full peer review and any attached files.

Reviewer #1: No

Reviewer #2: No

---

## [Editor Report · Decision Letter 1]

30 Sep 2019

Dear Dr. Maury,

We are pleased to inform you that your manuscript, "IL-4/IL-13 polarization of macrophages enhances Ebola virus glycoprotein-dependent infection", has been editorially accepted for publication at PLOS Neglected Tropical Diseases.

Before your manuscript can be formally accepted and sent to production you will need to complete our formatting changes, which you will receive in a follow up email. Please note: your manuscript will not be scheduled for publication until you have made the required changes.

IMPORTANT NOTES

* Copyediting and Author Proofs: To ensure prompt publication, your manuscript will NOT be subject to detailed copyediting and you will NOT receive a typeset proof for review. The corresponding author will have one final opportunity to correct any errors when sent the requests mentioned above. Please review this version of your manuscript for any errors.

* If you or your institution will be preparing press materials for this manuscript, please inform our press team in advance at plosntds@plos.org. If you need to know your paper's publication date for media purposes, you must coordinate with our press team, and your manuscript will remain under a strict press embargo until the publication date and time. PLOS NTDs may choose to issue a press release for your article. If there is anything that the journal should know, please get in touch.

*Now that your manuscript has been provisionally accepted, please log into EM and update your profile. Go to http://www.editorialmanager.com/pntd, log in, and click on the "Update My Information" link at the top of the page. Please update your user information to ensure an efficient production and billing process.

*Note to LaTeX users only - Our staff will ask you to upload a TEX file in addition to the PDF before the paper can be sent to typesetting, so please carefully review our Latex Guidelines [http://www.plosntds.org/static/latexGuidelines.action] in the meantime.

Best regards,

Thomas Geisbert

Associate Editor

Scott Weaver

Deputy Editor

---

## [Editor Report · Acceptance letter]

21 Nov 2019

Dear Dr. Maury,

We are delighted to inform you that your manuscript, "IL-4/IL-13 polarization of macrophages enhances Ebola virus glycoprotein-dependent infection," has been formally accepted for publication in PLOS Neglected Tropical Diseases.

Best regards,

Serap Aksoy

Editor-in-Chief

Shaden Kamhawi

Editor-in-Chief
